# Stiffness-Oriented Structure Topology Optimization for Hinge-Free Compliant Mechanisms Design

**Jincheng Guo** [1,2,*] and **Huaping Tang** [1,2]

1    School of Mechanical and Electrical Engineering, Central South University, Changsha 410083, China; huapingt-csu@163.com
2    State Key Laboratory of High-Performance Complex Manufacturing, Central South University, Changsha 410083, China
*    Correspondence: jcguo925@gmail.com; Tel.: +65-8224-9257

**Abstract:** This paper presents a stiffness-oriented structure topology optimization (TO) method for the design of a continuous, hinge-free compliant mechanism (CM). A synthesis formulation is developed to maximize the mechanism's mutual potential energy (MPE) to achieve required structure flexibility while maximizing the desired stiffness to withstand the loads. Different from the general approach of maximizing the overall stiffness of the structure, the proposed approach can contribute to guiding the optimization process focus on the desired stiffness in a specified direction by weighting the related eigen-frequency of the corresponding eigenmode. The benefit from this is that we can make full use of the material in micro-level compliant mechanism designs. The single-node connected hinge issue which often happened in optimized design can be precluded by introducing the eigen-frequency constraint into this synthesis formulation. Several obtained hinge-free designs illustrate the validity and robustness of the presented method and offer an alternative method for hinge-free compliant mechanism designs.

**Keywords:** stiffness-oriented; topology optimization; compliant mechanisms; hinge-free

## 1. Introduction

Distributed or hinge-free compliant mechanisms are continuous, monolithic (one-piece) and flexible mechanisms, which can transfer displacement and force from input port to output port in other directions by elastic deformation and have enough stiffness to withstand the external loads as well [1]. Compared with traditional rigid multi-part mechanisms, compliant mechanisms efficiently reduce the size, material usage, displacement loss, backlash loss, noise, and vibration [2,3]. Therefore, it shows considerable promise application in micro-scale and nano-scale mechanical systems such as biomedical devices, semiconductor industry, micro-electromechanical systems (MEMS), and many other applications. To obtain the compliant mechanisms, there are mainly two kinds of approaches, namely, the kinematics-based method and the structure topology optimization-based method.

Structure topology optimization has been successfully employed to achieve optimal topology design through various topology optimization algorithms, such as the homogenization algorithm [4], solid isotropic material with penalization (SIMP) algorithm [5], evolutionary structural optimization (ESO) algorithm [6,7], non-uniform rational basis spline (NURBS) based hyper-surfaces theory [8–10], and level-set [11] and phase-field approaches [12]. Numerical engineering examples for topology optimization validate its advantages in terms of systematism and high efficiency, compared with the kinematics-based approach. Therefore, the application of this technology has been further expanded to other fields over the last decades, such as compliant mechanisms design.

Compliant mechanisms are expected to achieve designated flexibility in the specified directions and to satisfy enough structure stiffness requirements to sustain external loads simultaneously. Accordingly, compliant mechanisms optimization can be categorized

to the multi-objectives problem using the topology optimization method. As a multi-objectives problem, one of the main difficulties in compliant mechanisms design is the provided formulation should properly tradeoff between flexibility and stiffness since there're conflicting design objectives with different fundamental units [13,14]. Shield and Prager [15] were the pioneers and proposed an energy-based formulation to unify these fundamental units by introducing mutual potential energy (MPE) to define flexibility requirement and strain energy (SE) to represent stiffness requirement. To maximize MPE and minimize SE simultaneously, Ananthasuresh et al. [16] presented a multi-criteria objective using a weighted linear combination of these two objectives. The difficulty is that the order of these two objectives is often not comparable and easy to result in one objective dominated by the other. A later approach by Frecker et al. [1] solved this problem using another multi-criteria formulation by maximizing the ratio of MPE and SE. Benefiting from that, the formulation successfully balanced the kinematics and structural stiffness. A further approach is presented by Saxena and Ananthasuresh [17,18] using a more general objective formulation by the ratio of $MPE^m$ and $SE^n$. Recently, Z. Luo et al. [19] proposed a new multi-objective formulation for topology optimization of compliant mechanisms which considers to maximum the MPE and minimum the mean compliance simultaneously.

Besides energy-based formulation, other approaches to design compliant mechanisms using topology optimization are studied to balance flexibility and stiffness requirements. An early study on topology synthesis of compliant mechanisms was carried out by Sigmund et al. [20] who introduced a formulation to maximize mechanical advantage (MA) with the volume and input displacement constraints. Along similar lines, Lau et al. [21] proposed a functional specification method by the ratio of geometry advantage (GA), MA to optimize the design of compliant mechanisms. To address the convex issue which may occur when regarding the maximum MA or GA as the objective function, Lau et al. [22] proposed a non-convex objective formulation and verified it with numerical examples. Pedersen et al. [23] introduced a material path-generating method and formulated an objective function for the synthesis design of large-displacement compliant mechanisms which was built by a global Lagrangian finite element formulation. In addition, Rubén Ansola et al. [24] introduced the ESO method to compliant mechanisms design by incorporating the elastic SE and MA into the objective function. Programing in this line of work, Rubén Ansola et al. [25] extended this method to the design of a 3D-compliant mechanism with a finite element addition scheme and introduced the elastic strain energy of the mechanism into the proposed objective function.

Apart from the difficulty to balance the flexibility and stiffness requirement in compliant mechanisms optimization formulation, the single-node hinge feature is another difficulty which frequently encounters in the design of compliant mechanisms using topology optimization, often leading to the optimized result close to a rigid-body mechanism [26]. The undesirable hinge feature is unfeasible in most practical application since it is very difficult to fabricate and easy to break due to its high-stress concentration; in other words, the single-node hinge feature often result in potential reliability issue. The optimization algorithm and formulation naturally lead to a single-node hinge feature, because it can generate large displacement under a given force while not increasing strain energy [27], therefore, worse local optima or non-optimal may occur in the topology optimization process. To achieve the design of hinge-free compliant mechanisms, Poulsen [28] cited a minimum length scale constraint in topology optimization and successfully generated distributed flexible compliant mechanisms. Furthermore, S. Rahmatalla et al. [29] developed a continuum structural topology optimization with hinge-free compliant mechanisms by attaching artificial springs to the input and output point of the optimization model and the single-node hinge zone was eliminated with an increase in the relative stiffness of the artificial springs. Recently, Y. Li et al. [2] introduced a new BESO algorithm to maximize the ratio of GA/SE, and the hinge zone was precluded by introducing the total strain energy of the structure.

In addition, the stiffness objective in general optimization formation is always an overall stiffness of the structure which is often achieved by the introduction of minimizing the SE into the formulation, but the actual applications are often required to focus on the stiffness in an interesting direction for optimizing the usage of material. Inspired by daily work on compliance mechanisms design, the method presented in this paper focuses on the stiffness in an interesting direction using eigenmodes by weighting its corresponding eigen-frequency. Because eigenmode is always related to specifically oriented stiffness and each eigenmode has its corresponding eigen-frequency, the optimization process will focus on the specified eigen-frequency when properly added into the objective formulation. Therefore, this paper develops a stiffness-oriented structure topological synthesis method to design a hinge-free compliant mechanism in which flexibility and stiffness requirements are considered. Based on the topology optimization and sensitivity analysis, a systematic formulation is deduced to tradeoffs between flexibility controlled by mutual potential and stiffness represented by eigen-frequency. Moreover, it will be shown that the unwanted single-node hinge connection issue has been successfully addressed by introducing the eigen-frequency constraints into the optimization formulation. This approach can provide an alternative method for hinge-free compliant design. Due to the output displacement and eigen-frequency can be efficiently monitored both in topology optimization results and practical industrial applications, the optimized design result can be easily reviewed. The presented topology optimization is based on SIMP interpolation and has been incorporated into a general finite element analysis software Hyperworks. Several examples are given to validate the effectiveness and robustness of the presented method.

The paper is outlined as follows. Section 2 presents a formulation named minimum synthesis weighting index (MSWI) in the framework of the SIMP-based method, moreover, checkerboard and minimum member size control are introduced as well. A sensitivity analysis of solutions to the MSWI has been derived in Section 3. The effectiveness of the proposed approach is proven through both 2D and 3D meaningful benchmarks in Section 4. Finally, a discussion on conclusions and future developments can be found in Section 5.

## 2. Complaint Mechanism Optimization Problem and Formulation

This section introduces a new topology optimization formulation that combines flexibility and stiffness requirement, conflicting design objectives of compliant mechanisms, into a single design objective.

### 2.1. Complaint Mechanism Optimization Problem

Consider a general compliance mechanism design example with design domain $\Omega$ under given loads and constraints conditions shown in Figure 1a where applied input force $F_A$ is the loading at the input port and results in a displacement $\Delta_A$. To achieve the flexibility requirement of compliant mechanisms, $\Delta_B$ is the expected output displacement at output port with a workpiece simulated by a spring with constant stiffness. Considering the stiffness requirement, the optimized design should be strong enough to withstand the input force $F_A$ and output reaction force $F_B$ from the workpiece.

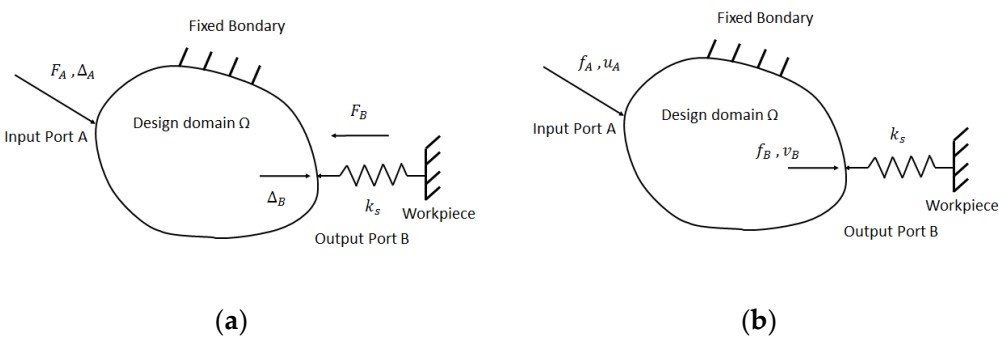

**Figure 1.** (**a**) Compliant mechanism design; (**b**) mutual potential energy (MPE) calculation.

To cater to the flexibility requirement, the compliant mechanism design is expected to be flexible enough to generate the required kinematic motion under the applied loads and reaction force from the workpiece. Generally, the structure flexibility can be represented by output displacement. Maximizing output displacement $\Delta_{\mathbf{B}}$ at port B can be expressed in terms of maximizing mutual potential energy(MPE), just as shown in Figure 1b, which can be formulated as follows [1].

$$\max\left(\mathbf{f}_B^T\mathbf{u}_A\right) = \max\left(\mathbf{v}_B^T\mathbf{K}_1\mathbf{u}_A\right) = \max(\text{MPE}) \tag{1}$$

Subject to:

$$\mathbf{K}_1\mathbf{u}_A = \mathbf{f}_A$$

$$\mathbf{K}_1\mathbf{v}_B = \mathbf{f}_B$$

where $\mathbf{u_A}$ is the nodal displacement vector due to $\mathbf{f}_A$, $\mathbf{v}_B$ is the nodal displacement vector due to the dummy load $\mathbf{f}_B$, and $\mathbf{K}_1$ is the symmetric global stiffness matrix.

On some occasions, optimized design layout often turns out to be connected with some single-node hinges which are very difficult to fabricate and easy to break due to their high-stress concentration and may result in potential reliability issues. The topology optimization algorithms prefer them because they can generate large displacements without increasing total strain energy. To deal with the hinge pivot issue in topology optimization, a general approach is introducing stiffness requirement into optimization formation as objective or constraint. Mostly, the structure stiffness is formulated in terms of the mean compliance or total strain energy [1,8,24]. In other words, maximizing the stiffness is equivalent to minimizing the mean compliance or total strain energy (SE).

However, the current optimization formulation cannot specifically focus on the stiffness in an interesting direction or eigenmode, because both mean compliance and total strain energy are scalar and cannot specify the direction. Generally, the mechanical structural stiffness can be characterized in terms of eigenfrequencies and their corresponding eigenmodes. It is a wise choice if the optimization process can focus on the interesting eigen-frequency and its corresponding eigenmodes. Another advantage of this approach is that eigen-frequency can be easily measured by various measurement methods, such as the sine sweeping-frequency vibration method and the hammering method. In addition, the single-node connected hinge can be precluded in compliant mechanism design by introducing the scheme of maximizing the required eigen-frequency. The reason is that material will be added to the hinge region to achieve the stiffness requirement.

The general eigenvalue problem can be formulated as follows,

$$\mathbf{K}\Phi_j = \omega_j^2\mathbf{M}\Phi_j = \lambda_j\mathbf{M}\Phi_j \tag{2}$$

where $\lambda_{\mathbf{j}} = \omega_j^2$ is the jth eigenvalue, $\omega_j$ is the jth eigen-frequency, $\mathbf{K}$ and $\mathbf{M}$ are the global stiffness and mass matrix respectively, $\Phi_{\mathbf{j}}$ is the corresponding eigenmode normalized with respect to the kinetic energy.

Assuming that damping can be neglected, the problem of maximizing a single eigenfrequency $\omega$ is equivalent to minimizing $1/\lambda$, which can be formulated as follows [30]:

$$\text{Min } \frac{1}{\lambda_j} \tag{3}$$

Subject to:

$$\mathbf{K}\Phi_j = \lambda_j\mathbf{M}\Phi_j, \; j = 1, \ldots, N,$$

$$\Phi_j^T\mathbf{M}\Phi_j = \delta_{jk}, \; j, k = 1, \ldots, N,$$

$$\sum_{e=1}^{N_e} \rho_e V_e - V^* \leq 0, \; V^* = aV_0$$

$$0 < \rho_{\min} < \rho_e < 1, \ e = 1, \dots, N_e$$

where $\mathbf{N}$ is the total number of eigenmodes related to the topology optimization and $\mathbf{N_e}$ is the number of elements. The symbol $\alpha$ is the volume fraction $V^*/V_0$, where $V_0$ is the admissible volume of the design domain and $V^*$ is the given available volume of design material. j and k are the eigenmode number. The density $\rho_e$ is the design variable, to avoid singularity happening, the minimal value of the design variable $\rho_{\min} = 10^{-3}$. $k_e$ and $m_e$ are the stiffness and the mass matrices with penalization factors p and q, respectively.

The globe stiffness matrix $\mathbf{K}$ which is assembled by the element stiffness matrix with design variables in the finite element structural analysis can be calculated as:

$$\mathbf{K} = \sum_{e=1}^{N_e} k_e = \sum_{e=1}^{N_e} \rho_e^p \mathbf{K}_e^0 \tag{4}$$

where $\mathbf{K}_e^0$ is the element stiffness matrix with fully solid material and penalization factor p $\geq \mathbf{1}$. In this paper, all the examples p = 3.

Similarly, the globe mass matrix $\mathbf{M}$ can be presented as follows:

$$\mathbf{M} = \sum_{e=1}^{N_e} m_e = \sum_{e=1}^{N_e} \rho_e^q \mathbf{M}_e^0 \tag{5}$$

where $\mathbf{M}_e^0$ is the element mass matrix with fully solid material and penalization factor q $\geq \mathbf{1}$. In this paper, all the examples q = 1.

### 2.2. Optimization Formulation

In this work, we can mathematically formulate an optimization problem by finding the design of a hinge-free optimal compliant mechanism by combining these two objectives, minimum of the -MPE and minimum $1/\lambda$, for different physical characters, while satisfying the above-mentioned constraints.

Min synthesis weighting index (MSWI):

$$\text{SWI} = \frac{-\sum_{i=1}^n W_i \text{MPE}_i}{\text{MPE}_{\max}} + \lambda_{\min} \sum_{i=1}^n \frac{W_i}{\lambda_i}, \ i = 1, \dots, n \tag{6}$$

Subject to:

$$\mathbf{K}_1 \mathbf{u}_A = \mathbf{f}_A$$

$$\mathbf{K}_1 \mathbf{v}_B = \mathbf{f}_B$$

$$\mathbf{K}\Phi_j = \lambda_j \mathbf{M}\Phi_j, \ j = 1, \dots, N,$$

$$\Phi_j^T \mathbf{M}\Phi_j = \delta_{jk}, \ j, k = 1, \dots, N,$$

$$\sum_{e=1}^{N_e} \rho_e V_e - V^* \leq 0, \ V^* = aV_0$$

$$0 < \rho_{\min} < \rho_e < 1, \ e = 1, \dots, N_e$$

where n is the total case number, $W_i$ is the weighting coefficient of the ith case, $\text{MPE}_{\max}$ is the maximum MPE and $\lambda_{\min}$ is the minimum $\lambda$, respectively.

The weighting of $\text{MPE}_i$ can be defined by $\sum_{i=1}^n W_i \text{MPE}_i$ and the weighting of $1/\lambda$. can be defined by $\sum_{i=1}^n W_i/\lambda$, respectively.

The formulation can handle multi-load cases by introducing the weighting $\sum_{i=1}^n W_i \text{MPE}_i$. For most of the compliant mechanisms, there're one or two load cases. Simultaneously, the formulation can focus the optimization process on the eigen-frequency and its corresponding eigenmodes by weighting the required eigen-frequency in formulation $\sum_{i=1}^n W_i/\lambda$.

Furthermore, by introducing the weighting coefficient, the formulation can balance these two different physical characters $\text{MPE}_i$ and $1/\lambda$, because their values may differ

by several orders of magnitude. In other words, the difficulty that one objective often dominates in the optimization process can be tackled, which often occurs when using a weighted linear combination of the two objectives in a multi-criteria optimization approach.

*2.3. Checkerboard and Minimum Member Size Control*

A well-known problem that often happened in continuum topology optimization results is the so-called checkerboard phenomenon. Diaz and Sigmund [31] introduced that the stiffness of the checkerboard pattern is overestimated and stiffer than any real material. As a result, the checkerboard pattern is artificially preferred by the optimization algorithm. Different approaches have been developed to overcome the checkerboard issue, Sigmund and Peterson [32] have given a review of the regularization methods as perimeter constraint, sensitivity filtering, and density gradient constraints.

Based on the understanding that the reason for the checkerboard pattern is due to bad numerical modeling that overestimates the stiffness of checkerboards, Petersson and Sigmund [33] introduced a constraint on the local gradient of the slope of element densities for guaranteeing the accuracy of the finite element formulation. However, these additional linear constraints make this approach computationally prohibitive for practical applications. Based on the basic concept of a slope constraint, Zhou et al. [34] enforced the formula by an adaptive constraint strategy in the optimization algorithm that is similar to adding move limits and does not require any extra computational effort. In the meantime, this algorithm is given an opportunity that can control the minimum member size.

For achieving a half predetermine minimum member size of the radius $r_{\min} = D_{\min}/2$, $D_{\min}$ denotes the minimum member size, the slope constraint can be formulated for a general irregular finite element mesh as follows:

$$|\rho_i - \rho_k| \leq \frac{(1.0 - \rho_{\min})\text{dist}(i,k)}{r_{\min}} \tag{7}$$

where $\text{dist}(i,k)$ is the distance between adjacent elements i and k, and $k \in \Omega_i$ with $\Omega_i$ denoting the set of elements adjacent to element i. However, the above formulation still adds additional linear constraints and result in computationally prohibitive as well. The density slope constraints in (7) can be improved through enforcement of adaptive lower bounds on the density as follows:

$$\rho_i \geq \max\left[\mu, \rho_j - (1.0 - \rho_{\min})\text{dist}(i,k)/r_{\min}\right] \tag{8}$$

where $\mu$ is the lower limit of density (in this paper, all examples $\mu = 0.6$) and $\rho_j$ is the density of element j at previous iteration that has the highest density among all elements that are adjacent to element i

$$\rho_j = \max(\rho_k | k \in \Omega_i) \tag{9}$$

Formula (8) shows that the only works with the values of the box-constraints on the density $\rho_j$, which at the (i + 1)th iteration step are modified to restrict the variations in the design. It is easy to see that except for the negligible computation associated with the adjustment of the side constraints of the design variables.

## 3. Sensitivity Analysis

To guide the iteration process of the optimization algorithm, sensitivity analysis is necessary to determine which elements should be removed or kept to the next iteration. Sensitivity analysis can be achieved by deriving the objective formulation $f(\rho_e)$ with respect to the design variable. For most topology optimization algorithms, the design variable is material density $\rho_e$. Based on the SIMP material model, the finite element elasticity matrix can be expressed by material density $\rho_e (0 \leq \rho_e \leq 1)$ with a power $p(p \geq 1)$, as below

$$E_e(\rho_e) = \rho_e^p E_e^0 \tag{10}$$

where $\mathbf{E}_e^0$ is the element elasticity matrix with fully solid material. The introduction of penalization power p is to penalize the elements with intermediate density so that only the element with the density close to 1 will be left in the optimization result, benefitting from that, the optimization result becomes easy to interpret.

The sensitivity of MPE variables can be deduced by:

$$\frac{\partial \mathbf{K}}{\partial \rho_e} \mathbf{u}_A + \mathbf{v}_B^T \mathbf{K} \frac{\partial \mathbf{u}_A}{\partial \rho_e} \tag{11}$$

Differentiating both sides of $\mathbf{K}_1 \mathbf{v}_B = \mathbf{f}_B$, then substituting the result into the above equation, the sensitivity of MPE with respect to $\rho_e$ can be rewritten as:

$$\frac{\partial \mathrm{MPE}}{\partial \rho_e} = \mathbf{v}_B^T \frac{\partial \mathbf{K}}{\partial \rho_e} \mathbf{u}_A = p\rho_e^{(P-1)} \mathbf{v}_B^T \mathbf{K}_e^0 \mathbf{u}_A \tag{12}$$

Similarly, the sensitivity of $\lambda$ can be deduced by differentiating the equation with respect to $\rho_e$ and we obtain [35]:

$$\frac{\partial \lambda}{\partial \rho_e} = \varphi_j^T \left( p\rho_e^{(P-1)} K_e^0 - \lambda_j q\rho_e^{(q-1)} M_e^0 \right) \varphi_j \tag{13}$$

$$E = 1, 2, 3 \ldots, N_E.$$

Finally, the sensitivity of synthesis weighting index (SWI) with respect to $\rho_e$ can be assembled and formulated as:

$$\frac{\partial \mathrm{SWI}}{\partial \rho_E} = -\frac{p\rho_e^{(P-1)} \sum_{i=1}^{n} W_i \mathbf{v}_B^T \mathbf{K}_e^0 \mathbf{u}_A}{\mathrm{MPE}_{max}} - \lambda_{min} \sum_{i=1}^{n} \frac{W_i \varphi_j^T \left( p\rho_e^{(P-1)} \mathbf{K}_e^0 - \lambda_j q\rho_e^{(q-1)} \mathbf{M}_e^0 \right) \varphi_j}{\lambda_i^2} \tag{14}$$

$$i = 1, \ldots, n$$

## 4. Numerical Results

In this section, three typical compliant mechanisms benchmark examples are presented to demonstrate the validity and robustness of the introduced methodology. The first two compliant mechanisms also can be found in other literature. It will be shown that the proposed method can be employed to solve the difficult tradeoffs between flexibility and stiffness and to solve the single node hinge issue as well.

The first two examples are 2D problems and the first eigenmode is swing modal in the thickness direction. Optimization results often turn out to be connected with the single node hinge, therefore, for the first two examples, the synthesis weighting index (SWI) formulation in Section 3 is employed to obtain a hinge-free design by weighting the 1st eigen-frequency with weighting coefficient 10 and 100, respectively.

### 4.1. Gripper Mechanism

The first example demonstrated here is one of the gripper mechanisms design. The mechanism is to achieve the functionality that two opposite points at the output side move in the vertical direction to grip a workpiece or a cell when a horizontal force or displacement is applied at the input point. The gripper design domain $\Omega$ as shown in Figure 2a is a 400 mm × 400 mm rectangle with a 120 mm × 120 mm gap at the output side. It is discretized into 3600 4-node quadrilateral elements. Partial fixed support is defined at both top and bottom corners on the left edge. The input load $f_{in} = 100$ N is exerted in the left direction at the input point. The objective is to synthesize the design so that the mechanism can produce the expected gripping force $f_{out}$ or output displacement $\Delta_{out}$ to grip the workpiece which is simulated by a spring with constant stiffness $k_{out} = 500$ N/mm, furthermore, the design should be stiff enough to sustain the applied load and reaction force from workpieces as well. The material considered here is Aluminum with Young's modulus E = 73 GPa, Poisson's ratio v = 0.3, and density $\rho = 2750$ kg/m$^3$.

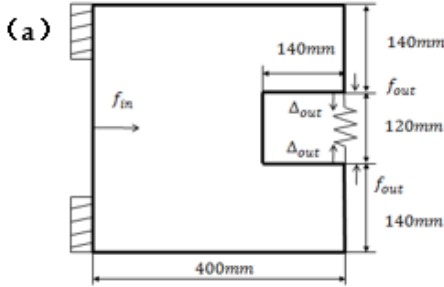

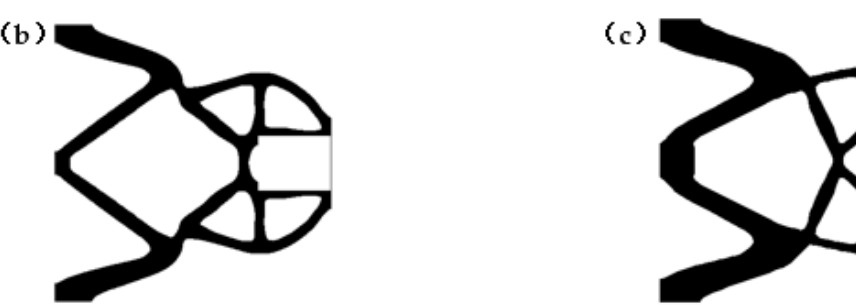

**Figure 2.** (**a**) Design domain and boundary of the gripper mechanism; (**b**) optimized topology solutions obtained by minimizing synthesis weighting index (SWI) with weighting coefficient 10 of the 1st eigen-frequency; (**c**) optimized topology solutions obtained by minimizing synthesis weighting index (SWI) with weighting coefficient 100 of the 1st eigen-frequency.

Figure 2b,c provides different optimized topology solutions which are well-distributed compliant designs with different weighting coefficients of the 1st eigen-frequency and the volume fraction constraint restricted to be 30%. The optimized topology design shows in Figure 2b a continuous, monolithic, flexible, and free of single-node hinge feature. When further increase the weighting coefficient of the 1st eigen-frequency, as Figure 2c shows the optimization process will switch to enhance the structural stiffness. Compared with the topology optimization result of References [19,29], the optimization result of Figure 2b,c basically shares a similar topology layout, but the topology layouts of Figure 2b,c are simpler and the load transmission paths are more concise and clear, which means these two designs will be easier to fabricate. The set of design results shown in Table 1 demonstrate that the weighting coefficient has a significant effect on the optimized topology design. The benefit of this is that the designer can conveniently obtain the expected optimized topology layout based on actual design requirements by weighting interested eigen-frequency.

**Table 1.** Two types of optimized result with different weighting coefficient of the 1st eigen-frequency.

| Case | Weighting Coefficient of the 1st Eigen-Frequency | Weighting Coefficient of the MPE | Output Stroke (μm) | The 1st Eigen-Frequency (Hz) | The 2nd Eigen-Frequency (Hz) |
|---|---|---|---|---|---|
| 1 | 10 | 1 | 10.6 | 733 | 2310 |
| 2 | 100 | 1 | 7.5 | 856 | 2581 |

Figure 3 illustrates the convergence history of output displacement and interested eigen-frequency of the gripper mechanism of case 1. Notice that the maximum displacement of 10.5 μm is achieved after iteration 35 with a table iteration process. Both 1st and 2nd eigen-frequency gradually increased until iteration 55 to achieve their maximum value. It shows that the optimization procedure focuses on the maximum interested eigen-frequency

after achieving the maximum MPE. The optimization results indicate that the proposed method works correctly.

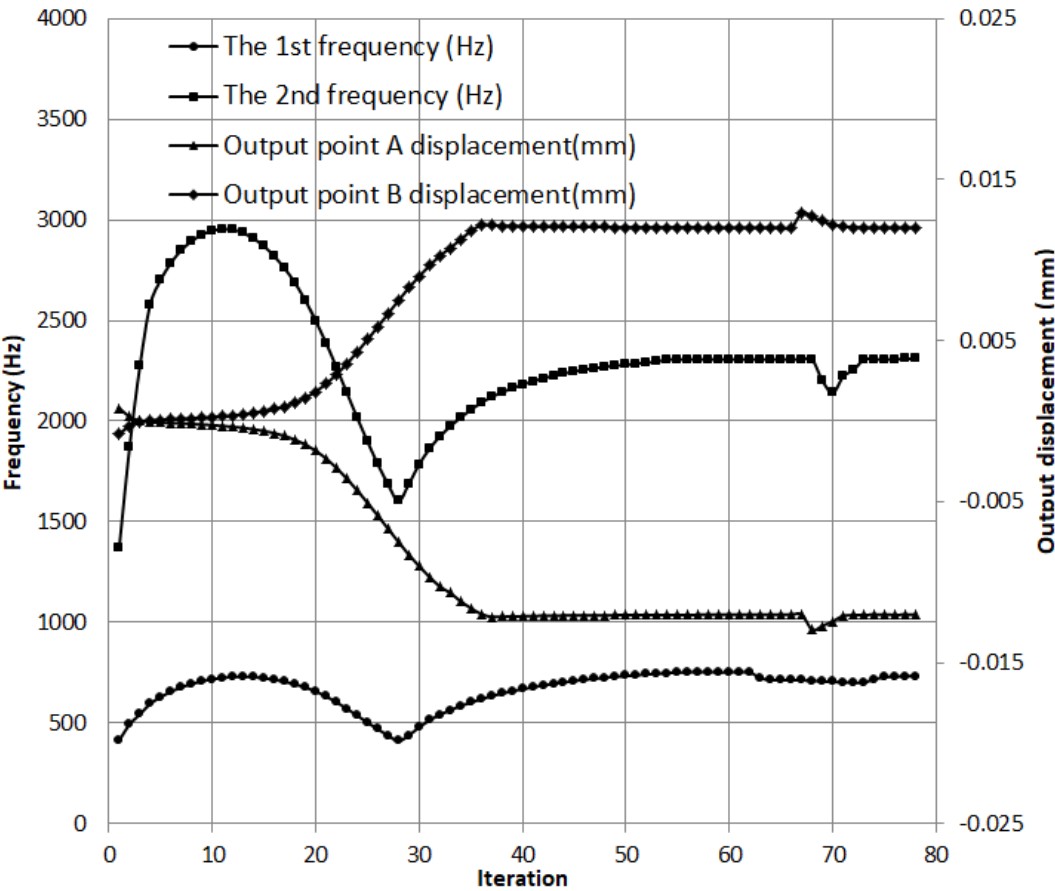

**Figure 3.** Convergence history of optimization for the gripper mechanism.

### 4.2. Inverter Mechanism

The second example demonstrates the design of a force or displacement inverter mechanism which outputs the displacement in an opposite direction to that of the input force. The design domain of the inverter is a 250 mm × 250 mm rectangle as shown in Figure 4a. It is discretized into 100 × 100 4-node bilinear quadrilateral elements. The design domain bears $F_{in} = 50$ N at the center of the left edge and is partially fixed at the top and bottom corners on the left edge. A linear spring with constant stiffness $k_{out} = 2000$ N/mm is attached to the output point to simulate the workpiece. The material is assumed to be Aluminum with Young's modulus E = 73 GPa, Poisson's ratio v = 0.3, and density $\rho = 2750$ kg/m$^3$. The horizontal displacement $\Delta_{out}$ of the output point is expected to be generated in the left direction and the design should be stiff enough to sustain the applied load and resistance force as well.

To highlight the effect of the varying weighting coefficient on optimized topology results. The problem was solved with different weighting coefficients of the 1st eigen-frequency as shown in Table 2. Figure 4b,c shows optimized designs for the inverter mechanism, clearly, the optimized layouts indicate that the optimized results are different but are continuous without any single-node hinge connection issue.

**Table 2.** Two types of optimized result with different weighting coefficient of the 1st eigen-frequency.

| Case | Weighting Coefficient of the 1st Eigen-Frequency | Weighting Coefficient of the MPE | Output Stroke (μm) | The 1st Eigen-Frequency (Hz) | The 2nd Eigen-Frequency (Hz) |
|------|------|------|------|------|------|
| 1 | 10 | 1 | 49.8 | 927 | 2480 |
| 2 | 100 | 1 | 32.4 | 1157 | 2613 |

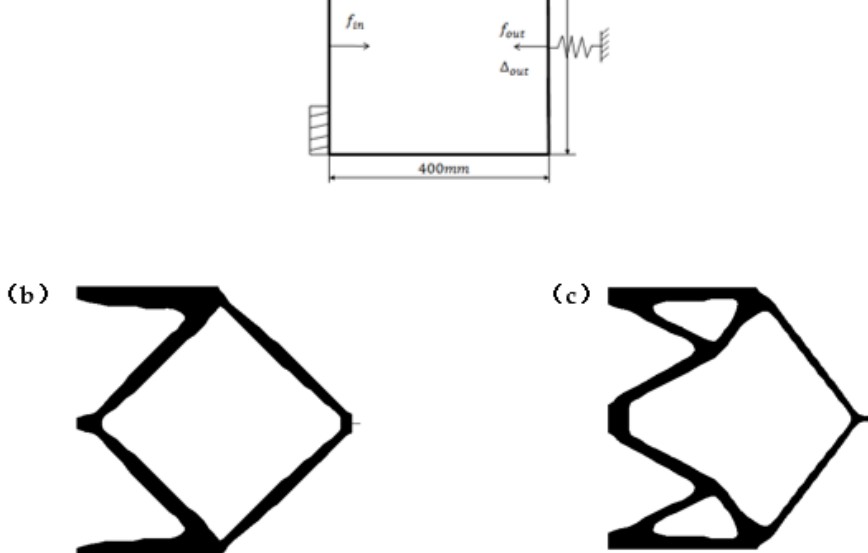

**Figure 4.** (**a**) Design domain and boundary of the inverter mechanism; (**b**) continuous optimized layout with weighting coefficient 10 of the 1st eigen-frequency; (**c**) continuous optimized layout with weighting coefficient 100 of the 1st eigen-frequency.

To further investigate the invert mechanism design, Figure 5 shows the convergence history of output displacement and interested eigen-frequency of the inverter mechanism of case 1. It clearly shows that the required output port moves in the desired direction and finally maximized the output stroke and the 1st eigenfrequencies, simultaneously.

### 4.3. Three-Dimensional Steering Mechanism

In this section, we consider the design of a 3D steering complaint mechanism using topology optimization to illustrate how to focus the optimization progress on interested eigen-frequency and its corresponding eigenmodes by proposed formulation and how to balance the flexibility and stiffness requirement by using weighting coefficient as well.

The mechanisms introduced here achieve the functionality that changes the input stroke in the horizontal direction, generated by PZT, to the output stroke in the vertical direction and amplify the input stroke simultaneously. The design domain $\Omega$ as shown in Figure 6a, a 40 mm × 25 mm × 15 mm block and another three blocks with gray color are non-design domains. The upper two small gray blocks attached on the upper surface are the adaptors of PZT and a bottom gray block regarded as a support beam. The whole design domain is discretized into 15,300 eight-node cubic elements. Partially fixed supports are defined at left two non-design blocks on the left side surfaces. The PZT, simulated as a spring with constant stiffness 20,000 N/mm, generated force $f_{in} = 200$ N is applied in the horizontal direction at the center of two PZT mounting surfaces. Similar to a cantilever mechanism, the first two eigenmodes are swing modal in the vertical and horizontal direction, respectively, as shown in Figure 6b,c. The first eigen-frequency requires more than 1500 Hz to achieve the required bandwidth for system motion control. The output

stroke should not be less than 50 um to achieve the flexibility requirement. Accordingly, the objective is to balance these contradictory requirements, the output stroke, and the 1st eigenfrequencies, simultaneously with a 20% volume constraint. For the elastic workpiece which is simulated as a spring, the constant stiffness of 500 N/mm is assigned. The material considered here is aluminum with Young's modulus $\mathbf{E} = 73$ GPa, Poisson's ratio $\mathbf{v} = 0.3$, and density $\rho = 2750$ kg/m$^3$.

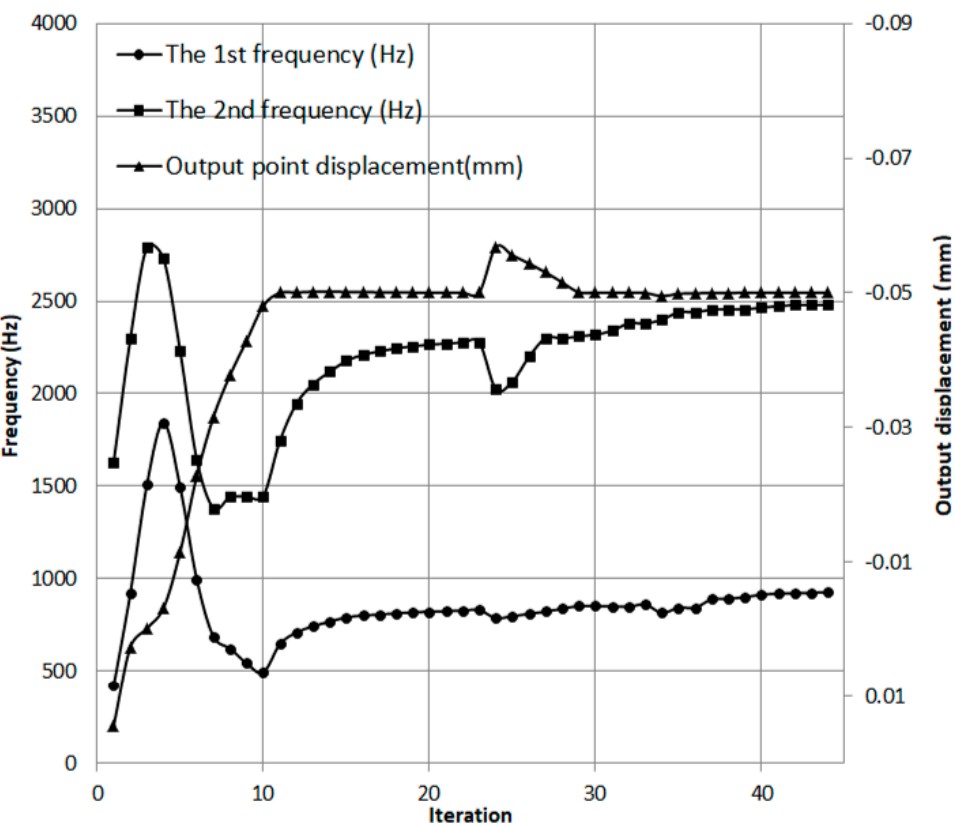

**Figure 5.** Convergence history of output displacement and eigen-frequency for the invent mechanism.

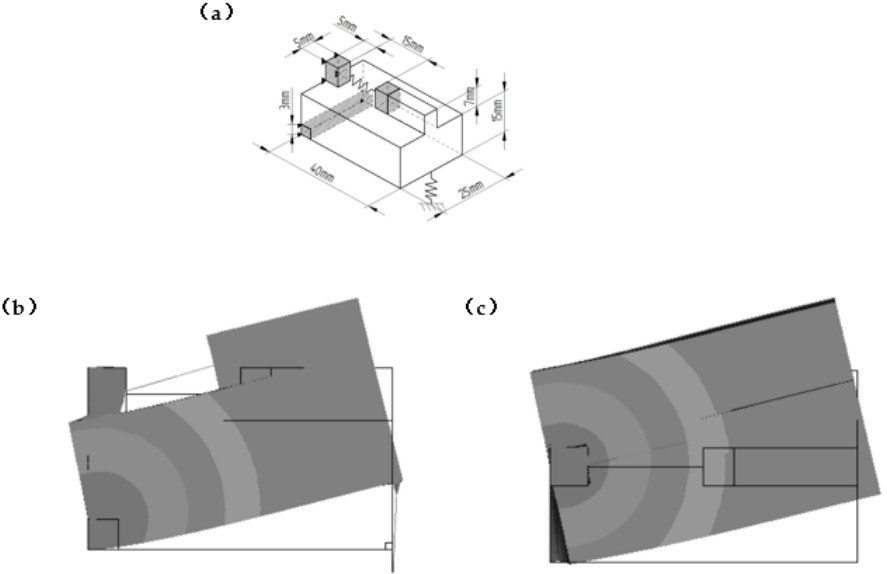

**Figure 6.** (**a**) Design domain and boundary of the steering mechanism; (**b**) the first eigenmode is a swing modal in vertical direction with eigen-frequency 1912 Hz; (**c**) the second eigenmode is a swing modal in horizontal direction with eigen-frequency 2186 Hz.

Clearly, the direction of the first eigenmode of the steering mechanism, swinging in the vertical direction, is the same as the required output stroke direction, as shown in Figure 6b. To fulfill the required the 1st eigen-frequency, not less than 1500 Hz, and output stroke, more than 50 µm, requirement simultaneously, the desired optimal result should properly balance the stiffness and flexibility requirements. Furthermore, to focus the optimization iteration process on the stiffness in the first eigenmode in the vertical direction to make full use of material, different weighting coefficients of the 1st eigen-frequency are employed as shown in Table 3.

**Table 3.** Four types of optimized result with different weighting coefficient of the 1st eigen-frequency.

| Case | 1 | 2 | 3 | 4 |
|---|---|---|---|---|
| Weighting coefficient of the 1st eigen-frequency | 1 | 100 | 1000 | 2250 |
| Weighting coefficient of the MPE | 1 | 1 | 1 | 1 |
| Output Stroke (µm) | 83 | 74 | 58 | 51 |
| The 1st eigen-frequency (Hz) | 1424 | 1721 | 2136 | 2404 |

Figure 7 shows four kinds of topology optimized results using different weighting coefficients of the 1st eigen-frequency from 1 to 2550. All of the optimized materials are well distributed and the topology layouts are continuous, monolithic, and free of single-node hinge features. The rotation pivots are close to the bottom non-design block and far away from the output point to achieve flexibility requirements. Based on Table 3, as the increasing weighting coefficient of the 1st eigen-frequency, the optimization iteration process will focus more on enhancing the 1st eigen-frequency, from 1424 Hz to 2404 Hz, by adding more material on rotation pivot while the output stroke reduced from 83 µm to 51 µm. Clearly, the topology optimized results vary with the weighting coefficient and the topology process will focus on the interested eigen-frequency with the help of the weighting coefficient.

Case 4 is chosen for further study of this steering mechanism, as Figure 8 shows, the optimal topology of this mechanism was obtained after 37 iterations using the proposed formulation. Figure 7g,h shows two different views of the optimized layout. The optimized design is a well-distributed compliant design and free of single-point hinge features. It clearly shows that the input force from PZT directly transmits to the output port to generate desired vertical stroke via the rotation pivot. There is a slanted bar connected between the output port and the main body to achieve the mentioned stiffness requirement. Figure 8 shows the convergence history of input/output strokes and the interested eigenfrequencies. The initial output stroke is 11 µm downwards and the final output stroke is 51 µm downwards which is larger than the required 50 µm requirement with the final GA of 4.6. The desired stiffness related to the first eigenmode swing in the vertical direction is enhanced and its corresponding eigen-frequency increased from 1252 Hz to 2404 Hz, which is also larger than the required 1500 Hz. Both of them indicate that the proposed method works correctly. Compared with the other three cases, case 4 optimized design added more material to pivot to enhance 1st eigen-frequency with the help of weighting efficiency. Finally, the 1st eigen-frequency 2404 HZ is achieved and increased the control bandwidth a lot.

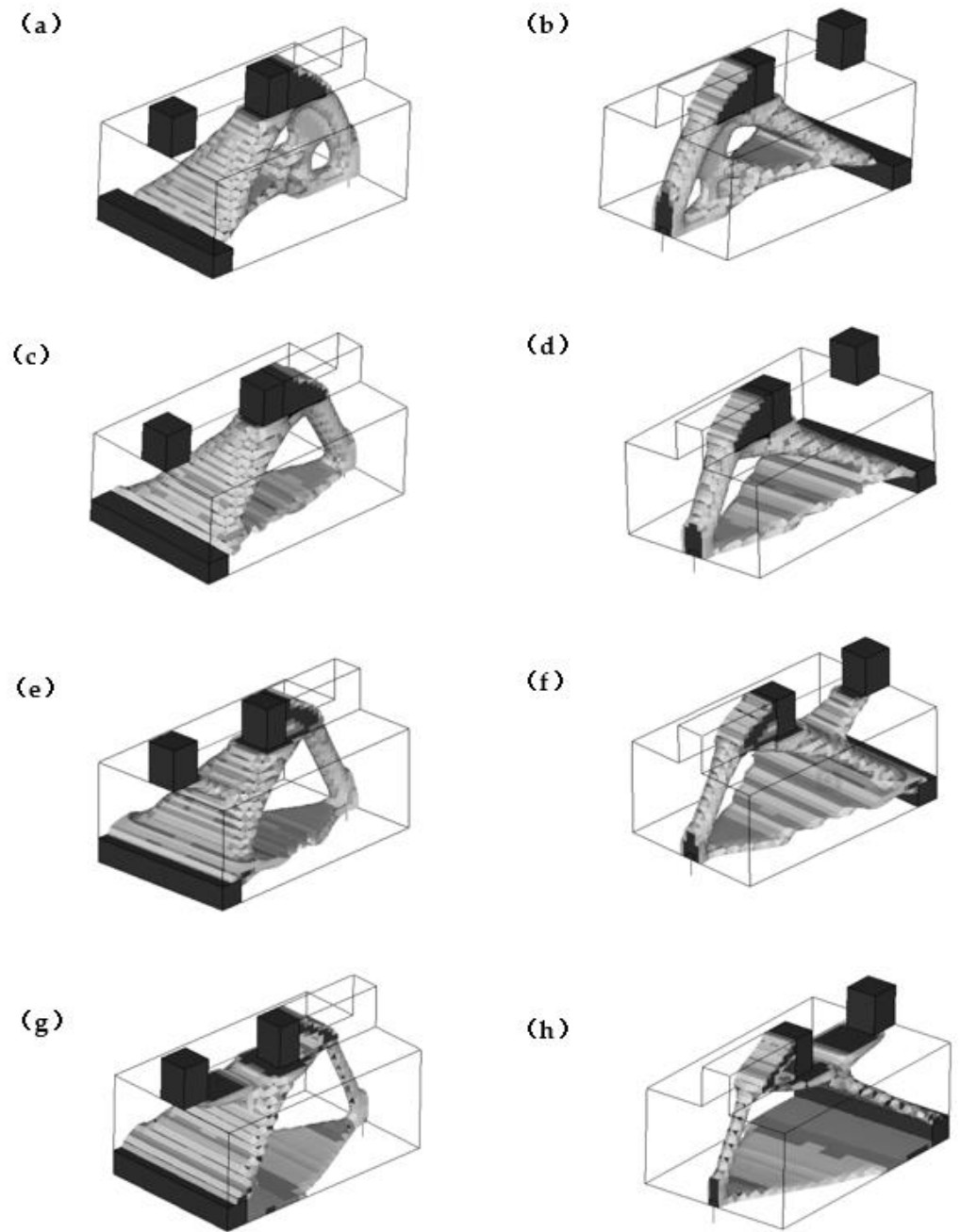

**Figure 7.** (**a**,**b**) continuous optimized layout  different views with weighting coefficient 1 of the 1st eigen-frequency; (**c**,**d**) continuous optimized layout different views with weighting coefficient 100 of the 1st eigen-frequency; (**e**,**f**) continuous optimized layout different views with weighting coefficient 1000 of the 1st eigen-frequency; (**g**,**h**) continuous optimized layout different views with weighting coefficient 2250 of the 1st eigen-frequency.

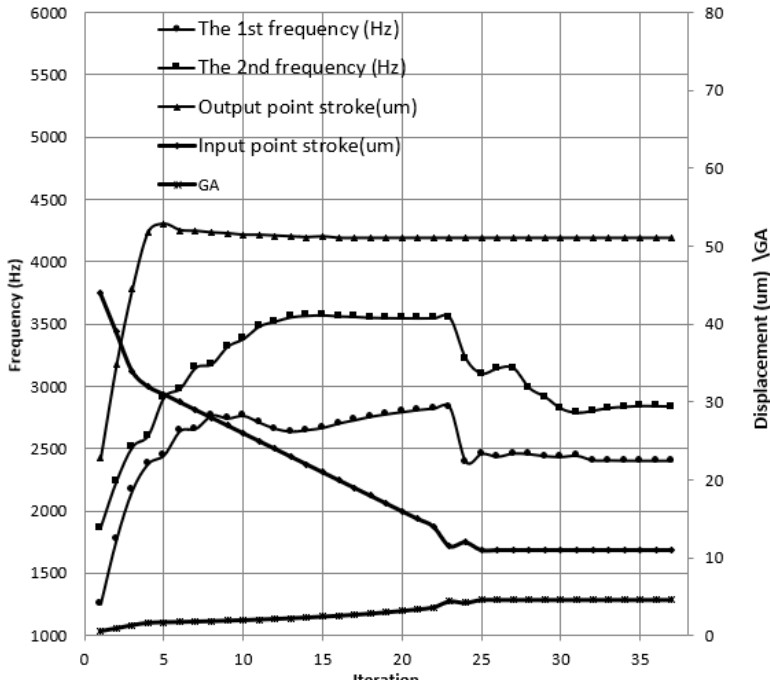

**Figure 8.** Convergence history of input and output strokes and eigenfrequencies for example 3 with weighting coefficient 2250 of the 1st eigen-frequency.

## 5. Conclusions

### 5.1. Conclusions

A stiffness-oriented structure topological synthesis method is proposed to design a hinge-free compliant mechanism, in which both flexibility and stiffness requirements are considered. The formulation of the compliant mechanism optimization problem presented here is to balance these two conflicting objectives simultaneously by combining the mutual potential energy and eigen-frequency into one objective. Numerically obtained hinge-free designs show that required output displacements are achieved through structure elastic deformation; meanwhile, the required stiffness of the specified eigenmode is maximized by weighting corresponding eigen-frequency, which demonstrates the validity of the presented method and offers an alternative method for hinge-free compliant mechanisms design. We notice that the proposed synthesis method can be easily extended to 3D problems or more complicated compliant problems, for instance, with the integration of a piezoelectric actuator for an actual compliant application.

### 5.2. Future Works

#### 5.2.1. Potential Spurious Modes and Mode Switching Problem

Spurious modes and mode switching problems often occur when introduced eigen-frequency into topology optimization formulation [9]. To overcome the possible spurious modes issues to achieve a more robust formulation, a more efficient penalization scheme on stiffness and mass matrix needs consider.

#### 5.2.2. Post-Processing Technology Improvement

To help designers efficiently make full use of optimized topology and easily transfer it to CAD environment for topology re-building and check the result with other CAE software, such as ANSYS, post-processing technology requires further improvements. For example, the use of CAD-compatible topology optimization methods [9–11].

**Author Contributions:** H.T. is responsible for the frame design of the paper and guide the writing of the paper; J.G. performed experiments and numerical studies on analytical objects and write the paper. All authors have read and agreed to the published version of the manuscript.

**Funding:** The research is supported by the Hunan Provincial Natural Science Foundation, Grant/Award Number: 2021JJ40735. The supports are gratefully acknowledged by the authors.

**Institutional Review Board Statement:** Not applicable.

**Informed Consent Statement:** Not applicable.

**Conflicts of Interest:** The authors declare no conflict of interest.

## Abbreviations

| | |
|---|---|
| TO | Topology optimization |
| MPE | Mutual potential energy |
| MEMS | Micro-electromechanical systems |
| SIMP | Solid isotropic material with penalization |
| ESO | Evolutionary structural optimization |
| ME | Compliant mechanisms |
| SE | Strain energy |
| GA | Geometry advantage |
| MA | Mechanical advantage |

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
