# Peer review of "Stiffness-Oriented Structure Topology Optimization for Hinge-Free Compliant Mechanisms Design"

_applsci, doi:10.3390/app112210831_

Round 1

Reviewer 1 Report

The article introduces a novel stiffness-oriented topology optimization (TO) formulation to design hinge-free compliant mechanisms. The investigated topic is surely worthy attention for the academic and industrial community and thus make this article attractive for interested readers. Moreover, the proposed numerical formulation is clearly presented and the conclusions are well supported by the provided numerical results.

However, in its current state the article is not suitable for publication yet. The following major and minor remarks should be addressed by the authors:

Major remarks:

  1. The provided reference list is rather short considering the high number of articles published on TO in general and compliant mechanisms design in particular. For instance, the author might consider the following references to extend the TO literature review, including different techniques rather than SIMP and ESO method, such as level-set[1] and phase-field approaches [2].
  2. From the given example it is not clear how the proposed algorithm differs from similar results already present in the literature. This point should be stated more clearly by the authors and they might consider adding a comparison with other results from the literature obtained using different approaches.

Minor remarks:

  1. 70-71; this sentence is quite unclear and should be rewritten.
  2. 237; why bold type?
  3. 294; um->µm, this should be modified in the entire text.
  4. 306; Figure4(a): space is missing, this error occurs several times in the text. Please, doublecheck.
  5. 307 edgeand: space is missing
  6. 334 4.3. but there is no 4.2 subsection.

References

[1]

H. Deng and A. C. To, "A Parametric Level Set Method for Topology Optimization Based on Deep Neural Network," Journal of Mechanical Design, 2021.

[2]

F. Auricchio, E. Bonetti, M. Carraturo, D. Hömberg, A. Reali and E. Rocca, "A phase-field-based graded-material topology optimization with stress constraint," Mathematical Models and Methods in Applied Sciences, vol. 30, no. 08, pp. 1461-1483, 2020.

Author Response

Dear Sir/Mdm,

    Thanks for your time to review my manuscript and thanks for your perfessional suggestions, which contribute a lot to my manuscript. 

  kindly help to refer to attachment for the detail response. 

Reviewer 2 Report

This work deals with the topology optimisation of compliant mechanisms by introducing a cost function based on a weighted sum of two requirements: the first one is related to the so-called mutual potential energy (MPE), while the second one is related to the first natural frequency. The problem formulation is completed by introducing an optimisation constraint on the volume fraction and the bounds on the design variables. The problem is solved by means of a classical density-based topology optimisation algorithm making use of the well-known solid isotropic material with penalisation (SIMP) penalisation scheme. The effectiveness of the approach is tested on 2D and 3D benchmark problems taken from the literature.

The topic could be of interest for the audience of Applied Sciences (APPSCI). Unfortunately, this work is characterised by a significant number of weaknesses, which make it not suitable for publication in the current form. Major revision is then recommended according to the comments listed here below.

1) The form. English should be improved throughout the manuscript. The form is often too colloquial. The style is often too verbose. Several typographical errors can be found in each section of the paper. Moreover, the English should be revised by an expert, native speaking. In the current form the manuscript is unreadable.

-The punctuation must be revised throughout the paper, especially for equations.

-The notation is not rigorous at all. Italic font should be used only for unknown scalar quantities. Vector and matrices should be indicated by means of lowercase and uppercase bold symbols and letters, respectively (not italic). Constant scalar parameters must be indicated without using italic font.

- A list of Acronyms should be provided for the sake of clarity, according to APPSCI format.

- Please, avoid the use of acronyms in the title of sections and subsections.

- Please, add the paper outline at the end of Introduction by concisely describe the content of each section (one/two sentences per section).

- There are numerous typos in the manuscript. A careful proofreading is needed to weed out typos

2) Introduction. The state of the art on modern and efficient density-based topology optimisation methods is not exhaustive at all. Authors are invited to quote and comment some of the following references:

[R1] NURBS Hypersurfaces for 3D Topology Optimisation Problems. Mechanics of Advanced Materials and Structures, v. 28 (7), pp. 665-684, 2021.

[R2] IgaTop: an implementation of topology optimization for structures using IGA in MATLAB. Structural and Multidisciplinary Optimization volume 64, pages1669–1700 (2021)

[R3] Structural Displacement Requirement in a Topology Optimization Algorithm Based on Isogeometric Entities. Journal of Optimization Theory and Applications, v. 184, pp. 250-276, 2020

[R4] A Comprehensive Review of Isogeometric Topology Optimization: Methods, Applications and Prospects. Chinese Journal of Mechanical Engineering volume 33, Article number: 87 (2020)

[R5] A NURBS-based Multi-Material Interpolation (N-MMI) for isogeometric topology optimization of structures. Applied Mathematical Modelling, Volume 81, May 2020, Pages 818-843

[R6] Eigen-frequencies and harmonic responses in topology optimisation: a CAD-compatible algorithm. Engineering Structures, v. 214, pp. 110602, 2020.

3) Section 2, lines 164-166: “In addition, the single-node connected hinge can be precluded in compliant mechanism design by introducing the scheme of maximizing required eigenfrequency”. To avoid the single-node connected hinge, the authors introduce a term related to the first natural frequency in the expression of the objective function. However, as discussed in [R6] (and in the main references quoted within [R6]) when dealing with eigen-frequencies and harmonic responses a particular care should be put in the problem formulation and in the penalisation law used for mass and stiffness matrices of the element. In particular, two main issues affect topology optimisation problems including structural responses based on eigen-frequencies: a) the spurious modes phenomenon; b) the mode switching phenomenon. The former can be avoided using a suitable penalisation law for both stiffness and mass matrices, whilst the latter can be reduced by introducing a suitable merit function depending upon a linear combination of the first N eigenfrequencies. The authors should agree that the problem formulation proposed in this study and the penalisation law used for stiffness and mass matrices do not allow avoiding these issues. In particular, the penalisation laws provided in Eqs. (4) and (5) do not allow avoiding spurious modes, whilst the objective function provided in Eq. (6) does not allow avoiding mode switching. Authors are invited to consider a more robust and efficient penalisation scheme and problem formulation, like those presented in [R6], and repeat again the numerical analyses for both 2D and 3D benchmarks problem. Otherwise, they are invited to explicitly admit the limitations of the proposed strategy and to add more results (in terms of the modes shapes related to the first N=6 eigen-frequencies, the evolution of the first N=6 along the iteration of the optimisation process) to clearly show that the optimised topologies illustrated in Sec. 4 are affected by both spurious modes and mode switching phenomena.

4) What is the value of parameter q used in Eq. (5) for each benchmark problem? Please provide more details on this point.

5) What is the value of the threshold density used at the end of the optimisation analysis to define the boundary of the optimised topology? How is it possible to recover the boundary of the optimised topology and to ensure an easy transfer to CAD environment for further post-processing and checking of results? Please, provide a discussion of the post-processing of the optimised topology and highlight the advantages and drawbacks of classical density-based topology optimisation method when compared to CAD-compatible topology optimisation methods like those discussed in [R1-R6].

6) What is the filtering technique used in this paper? How is checkerboard effect avoided? Please, provide more details and equations to explain this aspect.

7) What is the minimum member size used in each benchmark? How is it possible to control this feature in the proposed method?

8) The optimised topologies presented in this study are strongly affected by the values of the weights involved in the definition of the objective function. This is one of the main weaknesses of the proposed approach that should be clearly highlight in the manuscript. Please, revise accordingly.

9) The mesh of the finite element model used for the 3D benchmark problem is too coarse. Please, make use of a finer mesh and update the optimised topology.

Kind regards,

Author Response

Dear Sir/Mdm,

    Thanks for your time to review my manuscript and thanks for your perfessional suggestions, which contribute a lot to my manuscript. 

  kindly help to refer to attachment for the detail response. I already sent my revised manuscript to editor(Ms. Lili Chen), you can look for her if you want to review it. Thanks again. 

Round 2

Reviewer 1 Report

The authors addressed all the reviewers comment and thus the article is now suitable for publication.

Reviewer 2 Report

The authors provided a satisfactory reply to the comments raised by this reviewer. However the English grammar needs to be carefully checked because several typos are still present in the manuscript. Moreover, the state of the art about modern CAD-compatible topology optimisation methods should be improved. Anyway, the revised version of the paper can be considered for publication in Applied Sciences provided the above minor modifications are carried out.